# ET-SEED: Efficient Trajectory-Level SE(3) Equivariant Diffusion Policy

**Chenrui Tie**[1,2*]   **Yue Chen**[1*]   **Ruihai Wu**[1*]
**Boxuan Dong**[1]   **Zeyi Li**[1]   **Chongkai Gao**[2†]   **Hao Dong**[1†]
[1]Peking University   [2]National University of Singapore
chenrui.tie@u.nus.edu   yuechen@stu.pku.edu.cn   wuruihai@pku.edu.cn

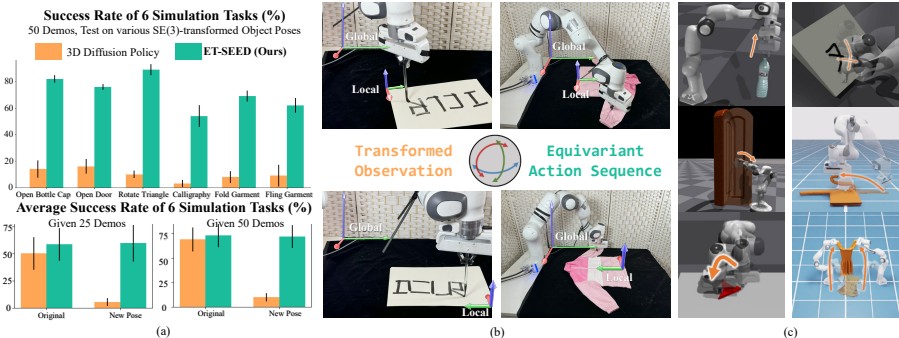

Figure 1: ET-SEED is a visual imitation learning algorithm that marries $SE(3)$ equivariant visual representations with diffusion policies. (a) ET-SEED achieve surprising **efficiency** and **spatial generalization** than baselines. (b) When the input object observation is rotated or translated, the output action sequence change equivariantly. (c) Visualizations of simulation environments.

## ABSTRACT

Imitation learning, *e.g.*, diffusion policy, has been proven effective in various robotic manipulation tasks. However, extensive demonstrations are required for policy robustness and generalization. To reduce the demonstration reliance, we leverage spatial symmetry and propose ET-SEED, an efficient trajectory-level $SE(3)$ equivariant diffusion model for generating action sequences in complex robot manipulation tasks. Further, previous equivariant diffusion models require the per-step equivariance in the Markov process, making it difficult to learn policy under such strong constraints. We theoretically extend equivariant Markov kernels and simplify the condition of equivariant diffusion process, thereby significantly improving training efficiency for trajectory-level $SE(3)$ equivariant diffusion policy. We evaluate ET-SEED on representative robotic manipulation tasks, involving rigid body, articulated and deformable object. Experiments demonstrate superior data efficiency and manipulation proficiency of our proposed method, as well as its ability to generalize to unseen configurations with only a few demonstrations. Videos and code are available at: https://et-seed.github.io/

## 1 INTRODUCTION

Imitation learning has achieved promising results for acquiring robot manipulation skills (Zhu et al., 2023; Fu et al., 2024; Chi et al., 2023; Wang et al., 2025). Though, one of the main challenges of imitation learning is that it requires extensive demonstrations to learn a robust manipulation policy (Brohan et al., 2022; Liu et al., 2024; Mandlekar et al., 2021). Especially once the spatial pose of the object to be manipulated runs out of the demonstration distribution, the policy performance will easily decrease. Although some works seek to tackle these issues through data augmentation (Yu et al., 2023) or contrastive learning (Ma et al., 2024), they usually require task-specific knowledge or extra training, and without theoretical guarantee of spatial generalization ability.

---

*Equal contribution

†Corresponding author

Another promising idea is to leverage symmetry. Symmetry is ubiquitous in the physical world, and many manipulation tasks exhibit a specific type of symmetry known as **SE(3) Equivariance**. $SE(3)$ is a group consisting of 3D rigid transformations. For example, as shown in fig. 1(b), a real robot arm is required to write characters "ICLR" on a paper or fold a garment, when the pose of the paper or the garment changes, the manipulation trajectories of the end-effector should transform equivalently. Employing such symmetries into policy learning can not only improve the data efficiency but also increase the spatial generalization ability. Recent works on 3D manipulation have explored using SE(3) equivariance in the imitation learning process. Most of these works focus on equivariant pose estimation of the target object or end-effector (Ryu et al., 2024; Hu et al., 2024; Gao et al., 2024). Trajectory-level imitation learning has achieved state-of-the-art performances on diverse manipulation tasks (Chi et al., 2023; Yang et al., 2024b). By generating a whole manipulation trajectory, this kind of method is capable to tackle more complex manipulation task beyond pick-and-place. For trajectory-level equivariance, Equivariant Diffusion Policy (Wang et al., 2024) and Equibot (Yang et al., 2024a) propose equivariant diffusion process for robotic manipulation tasks.

However, previous trajectory-level diffusion models for robotic manipulation have two key limitations. First, to maintain equivariance throughout the diffusion process, these models assume that every transition step must preserve equivariance. As we will show in section 4.1, training neural networks with equivariance is more challenging than neural networks with invariance, requiring additional computational resources and leading to slower convergence. This design constrains the model's efficiency, making it hard for tackling complex long-horizon manipulation tasks. Second, these models define the diffusion process in Euclidean space, which is not a natural definition, and limits the expressiveness. Since the focus is on equivariant diffusion processes within the $SE(3)$ group, it is more natural to define both the diffused variables and the noise as elements of the $SE(3)$ group, which will lead to better convergence and multimodal distributions representation (Urain et al., 2023).

In this work, we propose **ET-SEED**, a new trajectory-level $SE(3)$ equivariant diffusion model for manipulation tasks. ET-SEED improves the sample efficiency and decreases the training difficulty by restricting the equivariant operations during the diffusion denoising process. We extend the equivaraint Markov process theory and prove that during the full denoising process, at least only one equivariant transition is required. Then, we integrate the diffusion process on $SE(3)$ manifold (Jiang et al., 2024) and $SE(3)$ transformers (Fuchs et al., 2020) to design a new trajectory-level equivariant diffusion model on $SE(3)$ space. In experiment, we evaluate our method on several common and representative manipulation tasks, including rigid body manipulation (rotate triangle, open bottle cap), articulated object manipulation (open door), long-horizon tasks (robot calligraphy), and deformable object manipulation (fold and fling garment). Experiments show our method outperforms SOTA methods in terms of data efficiency, manipulation proficiency and spatial generalization ability. Further, in real-world experiments, with only 20 demonstration trajectories, our method is able to generalize to unseen scenarios.

In summary, our contributions are mainly as followed:

- We propose ET-SEED, an efficient trajectory-level $SE(3)$ equivariant diffusion policy defined on $SE(3)$ manifold, which achieves a proficient and generalizable manipulation policy with only a few demonstrations.
- We extend the theory of equivariant diffusion processes and derive a novel $SE(3)$ equivariant diffusion process, for simplified modeling and inference.
- We extensively evaluate our method on standard robot manipulation tasks in both simulation and real-world settings, demonstrating its data efficiency, manipulation proficiency, and spatial generalization ability, significantly outperforming baseline methods.

## 2 RELATED WORK

### 2.1 LEVERAGING EQUIVARIANCE FOR ROBOTIC MANIPULATION

Previous research has demonstrated that leveraging symmetry or equivariance in 3D Euclidean space can improve spatial generalization in a variety of robotic manipulation tasks. Lim et al. (2024); Hu et al. (2024); Simeonov et al. (2022); Xue et al. (2023b); Gao et al. (2024) proposed

$SE(3)$ equivariant model for grasp pose prediction. Other works have also leveraged this symmetry in tasks such as part assembly (Wu et al., 2023; Scarpellini et al., 2024), object manipulation on desktop (Wang et al., 2024), articulated and deformable object manipulation (Yang et al., 2024b) and affordance learning (Chen et al., 2024). Most of these studies either focus solely on generating a single 6D pose or fail to guarantee end-to-end equivariance across the entire $SE(3)$ space. In this paper, our proposed method is capable of generating manipulation trajectories while theoretically maintaining end-to-end equivariance over the entire $SE(3)$ group.

## 2.2 EQUIVARIANT DIFFUSION MODELS

Diffusion models (Sohl-Dickstein et al., 2015; Ho et al., 2020) compose a powerful family of generative models that have proven effective in robotic manipulation tasks (Chi et al., 2023). Previous studies (Guan et al., 2023; Schneuing et al., 2022) have investigated the effectiveness of combining spatial equivariance in the diffusion process to increase data efficiency and improve the spatial generalization ability of the model. GeoDiff (Xu et al., 2022) gave a theoretical proof of $SE(3)$ equivariant Markov process. Diffusion-EDFs (Ryu et al., 2024) and Orbitgrasp (Hu et al., 2024) introduced $SE(3)$ equivariant diffusion processes for target grasp pose prediction, but lack the capability to generate entire manipulation trajectories. Wang et al. (2024) proposed an equivariant diffusion policy capable of addressing $SO(2)$ equivariant tasks. EquiBot (Yang et al., 2024a) extended equivariant diffusion policies to $SIM(3)$ transformations, with the assumption that every transition step in the diffusion process is equivariant, which demands a high training cost. We further discuss the conditions of $SE(3)$ equivariant diffusion process and prove that not each, but at least one equivariant step is required. Based on this condition, we propose a novel $SE(3)$ equivariant diffusion model achieving better performance than previous works.

## 2.3 DIFFUSION ON $SE(3)$ MANIFOLD

Most diffusion models define the diffusion process on pixel space (Ho et al., 2020) or 3D Euclidean space (Chi et al., 2023). Leach et al. (2022) introduces a denoising diffusion model on $SO(3)$ group. $SE(3)$-Diffusion Fields (Urain et al., 2023) suggests that in 6-DoF grasp pose generation scenarios, formulating the diffusion process in $SE(3)$ manifold provides better coverage and representation of multimodal distributions, resulting in improved sample efficiency and performance. Jiang et al. (2024) proposes a $SE(3)$ diffusion model for robust 6D object pose estimation. In this work, we introduce an equivariant diffusion model on $SE(3)$ manifold for robot manipulation, revealing the superiority of defining equivariant diffusion process on $SE(3)$ over Euclidean space.

## 3 PRELIMINARY BACKGROUND

**SE(3) Group and its Lie Algebra**. $SE(3)$ (Special Euclidian Group) is a group consisting of 3D rigid transformations. Each $SE(3)$ transformation can be represented as a $4 \times 4$ matrix (denoted as $T$), indicating linear transformation on homogeneous 4-vectors. Formally, $T = \begin{bmatrix} R & \mathbf{t} \\ \mathbf{0} & 1 \end{bmatrix}$, where $R \in \mathbb{R}^{3\times3}$ is a rotation matrix, $\mathbf{t} \in \mathbb{R}^3$ is a translation vector. The Lie algebra $\mathfrak{se}(3)$ is a linear 6D vector space corresponding to the tangent space of $SE(3)$. Each element of $\mathfrak{se}(3)$ is a 6D vector $\delta \in \mathbb{R}^6$. The mutual mapping between $SE(3)$ and $\mathfrak{se}(3)$ is achieved by the logarithm map $\mathrm{Log} : SE(3) \rightarrow \mathbb{R}^6$ and the exponential map $\mathrm{Exp} : \mathbb{R}^6 \rightarrow SE(3)$. More information about $SE(3)$ and $\mathfrak{se}(3)$ can be found in appendix A.

$SE(3)$ **Equivariant Function**. Generally, we call a function $f : \mathcal{X} \rightarrow \mathcal{Y}$ that maps elements from input space $\mathcal{X}$ to output space $\mathcal{Y}$ is equivariant to a group $G$ if there are group representations of $G$ on $\mathcal{X}$ and $\mathcal{Y}$ respectively denoted by $\rho^{\mathcal{X}}$ and $\rho^{\mathcal{Y}}$ such that $\forall_{g \in G} : \rho^{\mathcal{Y}}(g) \circ f = f \circ \rho^{\mathcal{X}}(g)$

In other words, the function $f$ commutes with representations of the group $G$. In this paper, we focus on $SE(3)$ group, and represent elements of $SE(3)$ as $4 \times 4$ matrices. As special cases of general equivariance, we can define $SE(3)$ equivariant and invariant functions as:

---

**Definition 1** $SE(3)$ *Equivariant Function*.
*A function $f : \mathcal{X} \to \mathcal{Y}$ is called $SE(3)$ equivariant if*

$$\forall_{T \in SE(3)} : T \circ f = f \circ T \qquad (1)$$

**Definition 2** $SE(3)$ *Invariant Function*.
*A function $f : \mathcal{X} \to \mathcal{Y}$ is called $SE(3)$ invariant if*

$$\forall_{T \in SE(3)} : f = f \circ T \qquad (2)$$

---

$SE(3)$ **Equivariant Trajectory**. In many robotic manipulation tasks, the trajectories of the manipulator show a certain symmetry. If the representation of a trajectory under certain coordinate frame is $SE(3)$ invariant, we call the trajectory as $SE(3)$ *Equivariant*. Formally, it can be defined as

---

**Definition 3** $SE(3)$ *Equivariant Trajectory*.
*A trajectory $\{s_i\}_{i=1}^n$ is called $SE(3)$ equivariant if exists a coordinate frame $\mathcal{A}$, such that for any transformation $T \in SE(3)$ applied on both the trajectory and the coordinate frame(denoted as $\{s_i'\}_{i=1}^n = T\{s_i\}_{i=1}^n$ and $\mathcal{A}' = T\mathcal{A}$), the representation of $\{s_i'\}_{i=1}^n$ by the basis of $\mathcal{A}'$ is same as the representation of $\{s_i\}_{i=1}^n$ by the basis of $\mathcal{A}$.*

---

This property means the trajectory is "attached" on a certain frame, and when the frame transforms, the trajectory transforms accordingly. In our experiments, we select 6 representative manipulation tasks with this symmetry. Further discussion can be found in appendix G .

## 4 METHOD

**Problem Formulation.** We formulate the problem as an imitation learning setting, aiming to learn a mapping from observation $\mathbf{O}$ to action sequence $A$, with some demonstrations from an expert. In our setting, the observation $\mathbf{O}$ is colored point clouds $\mathbf{P} = \{(x_1, c_1), \cdots, (x_N, c_N)\} \in \mathbb{R}^{N \times 6}$. The action is defined directly as the desired 6D pose $\mathbf{H} \in SE(3)$ of the end-effectors. So in our setting, the action sequence means to the trajectory of end-effectors. This experimental setup does not require additional input information, and the action definition is both intuitive and consistent with real robot control, making it applicable to a wide range of robotic manipulation tasks.

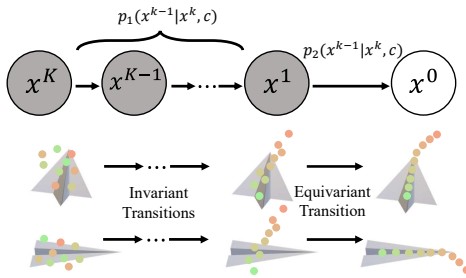

Figure 2: **Illustration of the denoising process of ET-SEED.** A random trajectory $x^K$ first passes through an invariant transition for $K - 1$ times and finally passes an equivariant transition once. This process is efficient while keeping $SE(3)$ equivariance property.

In this paper, we propose ET-SEED, a trajectory-level end-to-end $SE(3)$ equivariant diffusion model for robotic manipulation. **ET-SEED can theoretically guarantee the output actions are equivariant to any $SE(3)$ transformation applied on the input observation**, while only involving one equivariant denoising step. Fig. 2 is a general example to show how it works, given an observation and a noisy action sequence, our model first implement $K - 1$ invariant denoising steps, and pass the result into the last equivariant denoising step to generate a $SE(3)$ equivariant denoised trajectory.

We will discuss equivariant Markov processes further to explain the correctness and advantages of our proposed diffusion process in section 4.1 , with only one denoising step $SE(3)$ equivariant and the rest $SE(3)$ invariant. Then introduce our modified $SE(3)$ invariant and equivariant backbones in section 4.2 , and illustrate our $SE(3)$ equivariant diffusion process in section 4.3 .

### 4.1 EQUIVARIANT MARKOV PROCESS

For a Markov process $x^{K:0}$, and any roto-translational transformation $T \in SE(3)$. Geodiff (Xu et al., 2022) shows that if the initial probabilistic distribution is $SE(3)$ invariant, $i.e., p(x^K) = p(Tx^K)$, and the Markov transitions $p(x^{k-1}|x^k)$ are $SE(3)$ equivariant for any $1 \leq k \leq K$, $i.e., p(x^{k-1}|x^k) = p(Tx^{k-1}|Tx^k)$, then the density of $x_0$ satisfies $p(x_0) = p(Tx_0)$. Equibot (Yang et al., 2024a) adapts the theory and makes it more consistent with the robotics setting. They involve an additional condition $c$ (can be seen as an observation) and show that if the initial distribution $p(x^K|c)$ and transitions are all equivariant, $i.e., p(x^K|c) = p(Tx^K|Tc), p(x^{k-1}|x^k, c) = p(Tx^{k-1}|Tx^k, Tc)$ then the marginal distribution satisfies $p(x^0|c) = p(Tx^0|Tc)$.

In this paper, we discover that the condition of getting an equivariant marginal distribution $p(x^0|c)$ can be weaker. Formally, we first define three Markov transitions with different properties.

$$
\begin{aligned}
p_1(x^{k-1}|x^k, c) &= p_1(x^{k-1}|x^k, Tc) \\
p_2(x^{k-1}|x^k, c) &= p_2(Tx^{k-1}|x^k, Tc) \\
p_3(x^{k-1}|x^k, c) &= p_3(Tx^{k-1}|Tx^k, Tc)
\end{aligned}
\tag{3}
$$

Then we derive the marginal distribution using the three types of Markov transitions. We have the following statement. See appendix B for the detailed proof.

---

**Proposition 1** *For a Markov process $x^{K:0}$, if the initial distribution $p(x^K|c) = p(x^K|Tc)$, first $K - n + 1$ transitions follow the property of $p_1$, the middle 1 transition follows $p_2$, and the last $n - 2$ transitions follow $p_3$, then the final marginal distribution satisfies $p(x^0|c) = p(Tx^0|Tc)$.*

---

Previous works (Wang et al., 2024; Yang et al., 2024a) make all transitions $p_3$-like, which is a special case of proposition 1. In practice, we observe that training neural networks to approximate the properties of $p_2$ and $p_3$ is much more challenging compared to $p_1$, both in terms of performance and training cost. When the condition $c$ is transformed by a $SE(3)$ element, the distributions in $p_2$ and $p_3$ change equivalently, while the distribution in $p_1$ remains unchanged. Learning to output an equivariant feature is clearly more challenging for neural networks than producing an invariant feature. Additionally, in most of implementations of equivariant networks, building and training a model whose output is $SE(3)$ equivariant to the input takes up more computing resources than a $SE(3)$ invariant version. We design experiments to validate these facts. The results show that whether in single step or multiple steps setting, training invariant model is easier than equivariant model. Details of this confirmatory experiment can be found in appendix D.

In ET-SEED, we set the parameter $n = 2$, meaning there are $K - 1$ $p_1$-like transitions (referred to as "$SE(3)$ Invariant Denoising Steps") and one $p_2$-like transition (referred to as the "$SE(3)$ Equivariant Denoising Step"). This key design choice significantly reduces the training complexity, thereby enhancing the overall performance of our method.

### 4.2 $SE(3)$ EQUIVARIANT BACKBONE

In order to generate whole manipulation trajectories, it's necessary that the network has the ability to output a translation vector at anywhere in the 3D space (even beyond the convex hull of the object), which can not be achieved by directly using existing equivariant backbones (Fuchs et al., 2020; Deng et al., 2021; Liao & Smidt, 2022). In this paper, based on $SE(3)$ Transformer (Fuchs et al., 2020), we propose $SE(3)$ equivariant backbone $\mathcal{E}_{equiv}$ and invariant backbone $\mathcal{E}_{inv}$, which are suitable for predicting $SE(3)$ action sequences while theoretically satisfying definition 1 and 2. The implementation details can be found at appendix E . The input of backbone is a set of points coordinates $\mathcal{X} \in \mathbb{R}^{N \times 3}$, with some type-0 features $D_0^{in}$ and type-1 features $D_1^{in}$ attached on each point. Type-0 vectors are invariant under roto-translation transformations and type-1 vectors rotate and translate according to $SE(3)$ transformation of point cloud. The output can be elements of $SE(3)$ , each element is represented as a $4 \times 4$ matrix. For the $SE(3)$ equivariant model $\mathcal{E}_{equiv}$, we have

$$
\forall_{T \in SE(3)} : T\mathcal{E}_{equiv}(\mathcal{X}; D_0^{in}, D_1^{in}) = \mathcal{E}_{equiv}(T\mathcal{X}; D_0^{in}, TD_1^{in})
\tag{4}
$$

And for the $SE(3)$ invariant model $\mathcal{E}_{inv}$, we have

$$\forall_{T \in SE(3)} : \mathcal{E}_{inv}(\mathcal{X}; D_0^{in}, D_1^{in}) = \mathcal{E}_{inv}(T\mathcal{X}; D_0^{in}, TD_1^{in}) \tag{5}$$

---

**Algorithm 1** Training phase

**repeat**
    Sample $A^0, O \sim p_{data}$
    Sample $k \sim \text{Uniform}(\{1, ..., K\}), \varepsilon \sim \mathcal{N}(\mathbf{0}, \mathbf{I})$
    **for** $\mathbf{H}_i \in A^0$ **do**
        $\mathbf{H}_i^k = \text{Exp}\left(\gamma\sqrt{1 - \bar{\alpha}_t}\varepsilon\right) \mathcal{F}\left(\sqrt{\bar{\alpha}_t}; \mathbf{H}_i^0, \mathbb{H}\right)$
    Assign $A^k = \{\mathbf{H}_i^k\}_{i=0}^{T_p}$
    Predict $\hat{A}^{k \to 0} = s_\theta(O, A^k; k)$
    Optimize loss $\mathcal{L} = loss(\hat{A}^{k \to 0}, A^0(A^k)^{-1})$
**until** converged

**Algorithm 2** Inference phase

**for** $k = K, ..., 2$ **do**
    Predict $\hat{A}^{k \to 0} = \mathcal{E}_{inv}(O, A^k; k)$
    **for** $\mathbf{H}_i^k \in A^k$ **do**
        $\mathbf{H}_i^{k-1} = \text{Exp}(\lambda_0 \text{Log}(\hat{\mathbf{H}}_i^{k \to 0}\mathbf{H}_i^k)$
        $+ \lambda_1 \text{Log}(\mathbf{H}_i^k))$
    Assign $A^k = \{\mathbf{H}_i^k\}_{i=0}^{T_p}$
Predict $\hat{A}^{1 \to 0} = \mathcal{E}_{equiv}(O, A^1; 1)$
**for** $\mathbf{H}_i^1 \in A^1$ **do**
    Update $\mathbf{H}_i^0 = \hat{\mathbf{H}}_i^{1 \to 0}\mathbf{H}_i^1$
**Return**: $A^0 = \{\mathbf{H}_i^0\}_{i=0}^{T_p}$

---

### 4.3 $SE(3)$ EQUIVARIANT DIFFUSION PROCESS

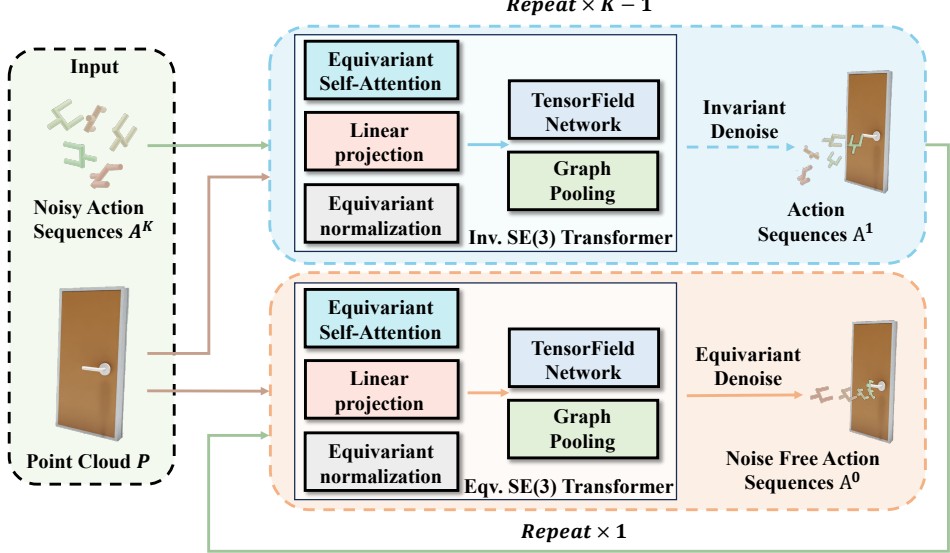

Figure 3: **Overview of our pipeline.** A colored point cloud and a random sampled action sequence are first passed through $K - 1$ $SE(3)$ invariant denoising steps and then a $SE(3)$ equivariant denoising step to generate a noise free action sequence. Although Inv. $SE(3)$ Transformer and Eqv. $SE(3)$ Transformer have same network architecture, the feature types of input and output differ, resulting in different coefficient matrices in network forward. Details can be referred to in appendix E.

Inspired by standard diffusion model, ET-SEED progressively disturbs the noise-free action $\mathbf{H}^0 \in SE(3)$ into a noisy action $\mathbf{H}^K$. As standard diffusion process assume the final noisy variable $x_T$ follows the standard Gaussian distribution $\mathcal{N}(\mathbf{0}, \mathbf{I})$, we assume the noisy action $\mathbf{H}^K$ follow a Gaussian distribution on $SE(3)$, centered at the identity transformation $\mathbb{H}$. So we use an interpolation-based $SE(3)$ diffusion formula, which represent the $\mathbf{H}^k \sim q(\mathbf{H}^k|\mathbf{H}^0)$ at noise step $k(1 \leq k \leq K)$ as

$$\mathbf{H}^k = \underbrace{\text{Exp}\left(\gamma\sqrt{1 - \bar{\alpha}_t}\varepsilon\right)}_{\text{Perturbation}} \underbrace{\mathcal{F}\left(\sqrt{\bar{\alpha}_t}; \mathbf{H}^0, \mathbb{H}\right)}_{\text{Interpolation}}, \varepsilon \sim \mathcal{N}(\mathbf{0}, \mathbf{I}) \tag{6}$$

The interpolation function $\mathcal{F}\left(\sqrt{\bar{\alpha}_t}; \mathbf{H}^0, \mathbb{H}\right)$ is an intermediate transformation between the origin action $\mathbf{H}^0$ and the identity transformation $\mathbb{H}$. By applying a perturbation noise $\text{Exp}\left(\gamma\sqrt{1 - \bar{\alpha}_t}\varepsilon\right)$ on the intermediate transformation, we get a diffused action $\mathbf{H}^k$. As shown in algorithm 1, we

design a model $s_\theta$ to predict the applied noise in a supervised learning fashion. More explanation about 6 can be found in Jiang et al. (2024) or appendix F.

This formulation is an analogy of DDPM (Ho et al., 2020), which represent the noisy image as

$$x_t = \bar{\alpha}_t x_0 + \bar{\beta}_t \bar{\varepsilon}, \bar{\varepsilon} \sim \mathcal{N}(\mathbf{0}, \mathbf{I}) \tag{7}$$

We can treat the first term of 7 as interpolation between $x_0$ and 0, second term as external noise.

The goal of $SE(3)$ reverse process is to train a denoising network, gradually refine the noisy action to the optimal ones. As illustrated in fig. 3 , the input of reverse process is an observation $O$, an noisy action sequence $A^K = [\mathbf{H}_0^K, \mathbf{H}_1^K, ..., \mathbf{H}_{T_p}^K]$, where $T_p$ is the action prediction horizon, and each $\mathbf{H}_i^K$ is drawn from a $SE(3)$ Gaussian distribution centered at identity transformation $\mathbb{H}$. The denoising process forms a Markov chain $A^K \rightarrow A^{K-1} \rightarrow \cdots \rightarrow A^0$. In each denoising step, the input of our denoising network $s_\theta$ consists of observation $O$, noisy action sequence $A^k$, and scalar condition $k$, outputs the predicted relative transformation between $A^k$ and noise-free action sequence $A^0$. Formally, we have

$$\hat{A}^{k \rightarrow 0} = s_\theta(O, A^k; k) \tag{8}$$

To ensure the overall $SE(3)$ equivariance of our pipeline, we propose a novel design of denoising network $s_\theta$. It consists of one $SE(3)$ invariant backbone $\mathcal{E}_{inv}$ and one $SE(3)$ equivariant backbone $\mathcal{E}_{equiv}$. We use $\mathcal{E}_{inv}$ to denoise in the first $K-1$ iterations and use $\mathcal{E}_{equiv}$ to denoise in the last iteration. Formally, $s_\theta$ is defined as

$$s_\theta(O, A^k; k) = \begin{cases} \mathcal{E}_{inv}(O, A^k; k), k > 1 \\ \mathcal{E}_{equiv}(O, A^k; k), k = 1 \end{cases} \tag{9}$$

As illustrated in algorithm 2 , in the first $K-1$ denoising iteration, we use $SE(3)$ invariant backbone $\mathcal{E}_{inv}$ to predict noise, and for each $\mathbf{H}_i^k$ in $A^k$, we use the $i$-th transformation $\hat{\mathbf{H}}_i^{k \rightarrow 0}$ of predicted noise sequence $\hat{A}^{k \rightarrow 0}$ to implement a denoise step by

$$\mathbf{H}_i^{k-1} = \mathrm{Exp}(\lambda_0 \mathrm{Log}(\hat{\mathbf{H}}_i^{k \rightarrow 0} \mathbf{H}_i^k) + \lambda_1 \mathrm{Log}(\mathbf{H}_i^k)) \tag{10}$$

This formulation, by minimizing the KL divergence of the posterior distribution and prior distribution of $\mathbf{H}_i^{k-1}$, is able to infer a more reliable distribution for $\mathbf{H}_i^{k-1}$ (Jiang et al., 2024).

By implementing the above $SE(3)$ invariant denoising step for $K-1$ times, we get the partially denoised action sequence $A^1$, which is invariant to any $SE(3)$ transformation of $O$. In the last denoising iteration, we use a $SE(3)$ equivariant backbone $\mathcal{E}_{equiv}$ to predict noise $\hat{A}^{1 \rightarrow 0}$ and directly apply each $\hat{\mathbf{H}}_i^{1 \rightarrow 0}$ of $\hat{A}^{1 \rightarrow 0}$ on corresponding action $\mathbf{H}_i^1$, i.e. $\mathbf{H}_i^0 = \hat{\mathbf{H}}_i^{1 \rightarrow 0} \mathbf{H}_i^1$. And finally get the noise-free action sequence $A^0 = [\mathbf{H}_0^0, \mathbf{H}_1^0, ..., \mathbf{H}_{T_p}^0]$.

With such design, the end-to-end equivariance is guaranteed. When the input observation $O$ is transformed by any $SE(3)$ element $T$, the output denoised action sequence $A^0$ will be equivariantly transformed. Formally, we have following proposition, with detailed proof attached in appendix C.

> **Proposition 2** *For a Markov process $A^{K:0}$, if $A^K$ is sampled from Gaussian distribution, $A^0 = ETSEED(A^K; O)$. Then, $\forall T \in SE(3) : TA^0 = ETSEED(TO, A^K)$.*

Table 1: Success rates (↑) and standard deviation of different tasks in simulation.

| | Open Bottle Cap | | | | Open Door | | | | Rotate Triangle | | | |
|---|---|---|---|---|---|---|---|---|---|---|---|---|
| | T | | NP | | T | | NP | | T | | NP | |
| Method | 25 | 50 | 25 | 50 | 25 | 50 | 25 | 50 | 25 | 50 | 25 | 50 |
| DP3 (Ze et al., 2024) | 65±4.5 | 76±5.5 | 11±4.2 | 14±6.5 | 61±2.24 | 72±2.74 | 9±3.54 | 16±5.48 | 67±2.74 | 89±2.24 | 5±2.24 | 10±2.74 |
| DP3+Aug | 35.0±5.0 | 44±4.2 | 38±4.47 | 46±7.42 | 43±8.37 | 54±6.52 | 30±4.18 | 40±8.22 | 35±3.54 | 42±4.47 | 32±5.70 | 41±4.18 |
| EquiBot (Yang et al., 2024a) | 63±2.74 | 73±2.74 | 63±5.70 | 77±7.58 | 56±2.24 | 72±2.24 | 58±7.58 | 77±7.58 | 67±2.74 | 84±2.24 | 64±8.66 | 86±5.48 |
| ET-SEED(Ours) | 67±2.74 | 81±2.24 | 74±6.52 | 82±2.74 | 66±2.24 | 75±2.74 | 66±2.74 | 76±2.24 | 83±2.24 | 93±2.74 | 85±2.24 | 89±4.18 |

| | Calligraphy | | | | Fold Garment | | | | Fling Garment | | | |
|---|---|---|---|---|---|---|---|---|---|---|---|---|
| | T | | NP | | T | | NP | | T | | NP | |
| Method | 25 | 50 | 25 | 50 | 25 | 50 | 25 | 50 | 25 | 50 | 25 | 50 |
| DP3 (Ze et al., 2024) | 28±2.74 | 50±3.54 | 0±0.00 | 3±2.74 | 44±2.24 | 60±4.18 | 4±5.48 | 8±4.48 | 36±5.48 | 67±4.48 | 4±5.48 | 9±8.22 |
| DP3+Aug | 8±2.74 | 21±4.18 | 3±2.24 | 12±11.51 | 13±5.70 | 27±7.58 | 17±10.37 | 31±9.62 | 28±7.58 | 38±4.48 | 11±4.18 | 31±2.24 |
| EquiBot (Yang et al., 2024a) | 24±5.48 | 43±8.37 | 14±10.84 | 40±10.61 | 34±4.18 | 58±2.74 | 33±2.74 | 60±7.90 | 35±6.12 | 61±6.52 | 36±6.52 | 64±8.22 |
| ET-SEED(Ours) | 38±2.74 | 55±3.54 | 36±6.52 | 54±8.22 | 47±2.74 | 67±2.74 | 49±2.24 | 69±4.18 | 50±5.00 | 67±4.48 | 48±4.47 | 62±5.70 |

Table 2: SE(3) Geodesic distances (↓) of different tasks in simulation.

| | Open bottle cap | | | | Open Door | | | | Rotate Triangle | | | |
|---|---|---|---|---|---|---|---|---|---|---|---|---|
| | T | | NP | | T | | NP | | T | | NP | |
| Method | 25 | 50 | 25 | 50 | 25 | 50 | 25 | 50 | 25 | 50 | 25 | 50 |
| DP3 (Ze et al., 2024) | 0.257 | 0.197 | 1.413 | 1.785 | 0.384 | 0.354 | 0.478 | 0.442 | 0.265 | 0.192 | 1.812 | 1.627 |
| DP3+Aug | 0.283 | 0.234 | 0.276 | 0.218 | 0.391 | 0.315 | 0.442 | 0.329 | 0.247 | 0.187 | 0.578 | 0.447 |
| EquiBot (Yang et al., 2024a) | 0.194 | 0.151 | 0.197 | 0.170 | 0.241 | 0.224 | 0.266 | 0.228 | 0.197 | 0.107 | 0.214 | 0.099 |
| ET-SEED (Ours) | 0.133 | 0.114 | 0.127 | 0.124 | 0.127 | 0.101 | 0.121 | 0.128 | 0.098 | 0.082 | 0.104 | 0.087 |

| | Calligraphy | | | | Fold Garment | | | | Fling Garment | | | |
|---|---|---|---|---|---|---|---|---|---|---|---|---|
| | T | | NP | | T | | NP | | T | | NP | |
| Method | 25 | 50 | 25 | 50 | 25 | 50 | 25 | 50 | 25 | 50 | 25 | 50 |
| DP3 (Ze et al., 2024) | 0.305 | 0.241 | 4.988 | 4.662 | 0.479 | 0.298 | 4.466 | 4.179 | 0.529 | 0.348 | 4.993 | 4.365 |
| DP3+Aug | 0.354 | 0.337 | 4.752 | 4.365 | 1.318 | 0.976 | 1.524 | 1.219 | 1.318 | 0.976 | 1.524 | 1.219 |
| EquiBot (Yang et al., 2024a) | 0.291 | 0.117 | 0.282 | 0.129 | 0.368 | 0.293 | 0.387 | 0.288 | 0.418 | 0.343 | 0.437 | 0.338 |
| ET-SEED (Ours) | 0.124 | 0.083 | 0.121 | 0.089 | 0.299 | 0.149 | 0.287 | 0.136 | 0.349 | 0.179 | 0.337 | 0.186 |

## 5 EXPERIMENTS

We systematically evaluate ET-SEED through both simulation and real-world experiments, aiming to address the following research questions: (1) Does our method demonstrate superior spatial generalization compared to existing imitation learning approaches? (2) Can our method achieve comparable performance with fewer demonstrations? (3) Is our method applicable to real-world robotic manipulation tasks?

### 5.1 SIMULATION EXPERIMENTS

**Tasks.** We design six representative robot manipulation tasks: *Open Bottle Cap*, *Open Door*, *Rotate Triangle*, *Calligraphy*, *Cloth Folding*, and *Cloth Fling*. These tasks encompass manipulation of rigid bodies, articulated bodies, and deformable objects, as well as dual-arm collaboration, long-horizon tasks, and complex manipulation scenarios. A brief overview is illustrated in fig. 1. For each task, we set up multiple cameras to capture full point clouds of the objects to be manipulated. We assume each robot manipulator operates within a complete 6DoF $SE(3)$ action space. Further details and discussions of their equivariant properties can be found in appendix G .

**Baselines.** We compare our method against the following baselines:

- **3D Diffusion Policy (DP3)** (Ze et al., 2024): A diffusion-based 3D visuomotor policy.

- **3D Diffusion Policy with Data Augmentations (DP3+Aug)**: Same architecture as DP3, with $SE(3)$ data augmentation added.

- **EquiBot** (Yang et al., 2024a): A baseline combines SIM(3)-equivariant neural network architectures with diffusion policy.

DP3 and DP3+Aug are used to compare ET-SEED with baseline methods that utilize data augmentation to achieve spatial generalization, while EquiBot allows for a comparison between different architectures of equivariant diffusion process.

**Augmentations.** The DP3+Aug baseline utilizes augmentations during training. In all environments, training data is augmented by (1) rotating the observation along all three axes by random angles between $0°$ and $90°$, and (2) applying a random Gaussian offset to the observation. The standard deviation of the Gaussian noise is set to 10% of the workspace size.

**Evaluation.** Following the setup of Gao et al. (2024), we collect demonstrations and train our policy under the **Training setting (T)**, subsequently testing the trained policy on both T and **New Poses (NP)**, where target object poses undergo random $SE(3)$ transformations. We evaluate all methods using two metrics, based on 20 evaluation rollouts, averaged over 5 random seeds. Since we generate complete manipulation trajectories, the final success rate alone is inadequate for fully assessing the trajectory's quality. We calculate the geodesic distance between each step of the predicted trajectory and the ground truth trajectory, providing a more comprehensive reflection of the trajectory's overall quality. The geodesic distance between each step of the predicted trajectory and the ground truth trajectory, we can obtain a more accurate reflection of the trajectory's overall quality. The definition of geodesic distance between $T, \hat{T} \in SE(3)$ is

$$\mathcal{D}_{geo}(\mathbf{T}, \hat{\mathbf{T}}) = \sqrt{\left\| \text{Log}(\mathbf{R}^T \hat{\mathbf{R}}) \right\|^2 + \left\| \hat{\mathbf{t}} - \mathbf{t} \right\|^2}, \tag{11}$$

where $R$ and $t$ are the rotation and translation parts of $T$. We report $\mathcal{D}_{geo}$ in the same manner as success rates.

**Results.** Table 1 and 2 provide a quantitative comparison between our method and the baseline. Both DP3 and its augmented variant demonstrate strong performance in the training setting (T), but they exhibit a significant drop in performance when faced with New Poses (NP) scenarios. This highlights that merely incorporating data augmentation is insufficient for the model to generalize effectively to unseen poses. Instead, leveraging equivariance proves essential for enhancing spatial generalization.

While EquiBot achieves commendable results in both success rate and $\mathcal{D}_{geo}$, it struggles with more complex, long-horizon tasks such as Calligraphy and Fold Garment. Also, when less demonstrations are given, the performance is not satisfactory. These challenges stem from the inherent complexity of its diffusion process design, where maintaining equivariance in each Markov transition adds substantial difficulty to the learning task.

Table 3: Ablation studies.

| Design | Average |
|---|---|
| Ours w/o SE(3) | 24±4.48 |
| Ours w/o Eqv-Diff | 57±6.52 |
| **Ours** | **76±2.24** |

In contrast, ET-SEED consistently outperforms across all six tasks, with minimal performance drop when facing unseen object poses. This advantage is especially pronounced when using a limited number of demonstrations, showcasing ET-SEED's superior data efficiency, manipulation proficiency, and spatial generalization ability.

**Ablation Studies.** We conduct ablation studies on the New Pose (NP) scenario of the representative *Opening Door* task to evaluate the effectiveness of different components of our approach:

- **Ours w/o SE(3)**: Our method without $SE(3)$ invariance and equivariance in the backbone architecture. In this variant, we use a standard PointNet++ to predict noise at each step.

- **Ours w/o Eqv-Diff**: Our method without the $SE(3)$ equivariant denoising process. Instead, we use a non-equivariant diffusion process (DDIM), following the approach of Ze et al. (2024).

Table 3 shows quantitative comparisons with ablations. Clearly each component improves our method's capability.

## 5.2 REAL-ROBOT EXPERIMENT

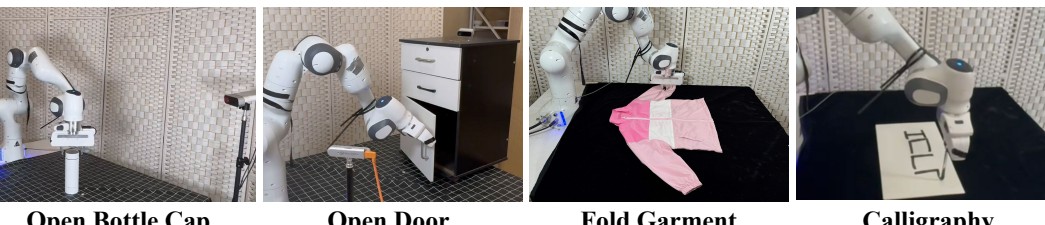

| **Open Bottle Cap** | **Open Door** | **Fold Garment** | **Calligraphy** |

Figure 4: Visualizations of the real-world environments used in our experiments. The tasks are performed using multiple Microsoft Azure Kinect cameras and Intel® RealSense for point cloud fusion and a Franka robotic arm for execution.

**Setup.** We test the performance of our model on four tasks on real scenarios. All the tasks are visualized in Figure 4 . We use Segment Anything Model 2 (SAM2) (Ravi et al., 2024) to segment the object from the scene and project the segmented image with depth to point cloud. Please refer to appendix H and our website for more details and videos of real-world manipulations.

Expert demonstrations are collected by human tele-operation. The Franka arm and the gripper are teleoperated by the keyboard. Since our tasks contain more than one stage and include two robots and various objects, making the process of demonstration collection very time-consuming, we only provide 20 demonstrations for each task.

In test setting, We place the object at 10 different positions with different poses that are unseen in the training data. Each position is evaluated with one trial.

**Results.** Results for our real robot tasks are given in Table 4. Consistent with our simulation findings, in real world experiments, ET-SEED performs better than baselines in all the four tasks, given only 20 demonstrations. The evaluation shows the effectiveness and spatial generalization ability of our method.

Table 4: Success rates in real-world robot experiments.

| Method | Open Bottle Cap | Open Door | Calligraphy | Fold Garment |
|---|---|---|---|---|
| DP3 | 0.2 | 0.2 | 0.0 | 0.1 |
| DP3+Aug | 0.2 | 0.3 | 0.0 | 0.2 |
| EquiBot | 0.6 | 0.5 | 0.0 | 0.3 |
| ET-SEED (Ours) | **0.8** | **0.6** | **0.4** | **0.6** |

## 6 CONCLUSION

In this paper, we propose ET-SEED, an efficient trajectory-level $SE(3)$ equivariant diffusion policy. Our method enhances both data efficiency and spatial generalization while reducing the training complexity typically encountered in diffusion-based methods. Through theoretical extensions of equivariant Markov kernels, we demonstrated that the $SE(3)$ equivariant diffusion process can be achieved given a weaker condition, significantly simplifying the learning task. Experimental results on diverse robotic manipulation tasks show that ET-SEED performs better than SOTA methods. Real-world experiments further validate the generalization ability of our model to unseen object poses with only 20 demonstrations. ET-SEED is a novel approach for data efficient and generalizable imitation learning, paving the way for more capable and adaptive robots in real-world applications. However, the proposed method has certain limitations. 1) The equivariance of ET-SEED is designed for object point clouds, so, pre-processing is required to extract segmented object point clouds from the scene point clouds. 2) The action space is defined as the 6D poses of the end-effectors, and we assume the end-effectors can reach any feasible 6D pose by inverse kinematics solvers and controllers. Additionally, this action space is not able to tackle dexterous manipulation tasks as they need action space with higher dimension.

## 7 ACKNOWLEDGMENT

We gratefully acknowledge the financial support provided by the National Youth Talent Support Program [grant number 8200800081] and the National Natural Science Foundation of China [grant number 62376006]. This funding has been instrumental in enabling our research and the successful completion of this work.

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

# A $SE(3)$ GROUP AND $\mathfrak{se}(3)$ ALGEBRA

In this section, we briefly introduce the $SE(3)$ Lie group and $\mathfrak{se}(3)$ Lie algebra. One can refer to Eade (2013) for more detailed explaination.

## A.1 BASIC TERMS

An element in $SE(3)$ can be represented as a $4 \times 4$ matrix

$$
\mathbf{R} \in SO(3), t \in \mathbb{R}^3
$$
$$
C = \begin{pmatrix} \mathbf{R} & t \\ 0 & 1 \end{pmatrix} \tag{12}
$$

This representation means we can compute the composition and inversion of elements in $SE(3)$ by matrix multiplication and inversion.

An element $\delta = se(3)$ can be represented by multiples of the generators

$$
\delta = (\mathbf{u}, \omega)^T \in \mathbb{R}^6 \tag{13}
$$

$\mathbf{u}$ is the translation, $\mathbf{u} \in \mathbb{R}^3$

$\omega$ is the rotation (exactly the axis-angle representation, its normal is the rotation angle, and its direction is the rotation axis), $\omega \in \mathbb{R}^3$

There's a 1-1 map between $SE(3)$ and $se(3)$

$$
\delta = ln(C)
$$
$$
C = exp(\delta) \tag{14}
$$

## A.2 INTERPOLATION

two elements $a, b \in G$, we would like to interpolate between the two elements according to a parameter $t \in [0, 1]$, define an interpolation function

$$
f : G \times G \times \mathbb{R} \to G \tag{15}
$$

First define a group element that tasks $a$ to $b$

$$
d := b \cdot a^{-1} d \cdot a = b \tag{16}
$$

Compute the corresponding Lie algebra vector and scale it by $t$

$\mathbf{d}(t) = t \cdot ln(d); d_t = exp(\mathbf{d}(t))$
$$
f(a, b, t) = d_t \cdot a \tag{17}
$$

## A.3 GAUSSIAN DISTRIBUTION

Consider a Lie group $G$ and its Lie algebra vector space $\mathfrak{g}$, with $k$ DoF. A mean transformation $\mu \in G$ and a covariance matrix $\Sigma \in \mathbb{R}^{k \times k}$. We can sample an element from the Gaussian distribution on $G$

$$
\delta \sim \mathcal{N}(0; \Sigma)(\delta \in \mathfrak{g})
$$
$$
x = exp(\delta) \cdot \mu \tag{18}
$$

c.f. Gaussian in $\mathbb{R}^D$

$$
N(\mathbf{x} \mid \mu, \mathbf{\Sigma}) = \frac{1}{(2\pi)^{D/2}} \frac{1}{|\mathbf{\Sigma}|^{1/2}} \exp\left\{ -\frac{1}{2}(\mathbf{x} - \mu)^T \mathbf{\Sigma}^{-1}(\mathbf{x} - \mu) \right\} \tag{19}
$$

## B  PROOF OF PROPOSITION 1

$$
p(x^0|c) = \int p(x^K|c)p(x^{0:K-1}|x^K,c)dx^{1:K}
$$

$$
= \int p(x^K|c)\prod_{k=1}^{K} p(x^{k-1}|x^k,c)dx^{1:K}
$$

$$
= \int p(x^K|c)(\prod_{k=n}^{K} p(x^{k-1}|x^k,c))p(x^{n-2}|x^{n-1},c)(\prod_{i=1}^{n-2} p(x^{i-1}|x^i,c))dx^{1:K}
$$

$$
= \int p(x^K|Tc)[\prod_{k=n}^{K} p_1(x^{k-1}|x^k,Tc)]p_2(Tx^{n-2}|x^{n-1},Tc)
$$

$$
[\prod_{i=1}^{n-2} p_3(Tx^{i-1}|Tx^i,Tc)]dx^{1:K}
$$

$$
= p(x^{n-1}|Tc)\, p(Tx^{n-2}|x^{n-1},Tc)\, p(Tx^0|Tx^{n-2},Tc)
$$

$$
= p(Tx^0|Tc)
$$

(20)

## C  PROOF  PROPOSITION 2

Here, for simplicity, we define two operations. 1) For a matrix $T$ and a list of matrix $A = [\mathbf{H}_1,...,\mathbf{H}_N]$, the notation $TA$ means multiplying $T$ on each $\mathbf{H}_i$. $i.e., TA = [T\mathbf{H}_1,...,T\mathbf{H}_N]$. 2) For two lists of matrix $A = [\mathbf{H}_1,...,\mathbf{H}_N]$, $B = [\mathbf{G}_1,...,\mathbf{G}_N]$, the notation $AB$ means multiplying the matrices at the corresponding positions. $i.e., AB = [\mathbf{H}_1\mathbf{G}_1,...,\mathbf{H}_N\mathbf{G}_N]$

As shown in 4 and 5, our backbones are theoretically $SE(3)$ equivariant and invariant, i.e.

$$
\mathcal{E}_{inv}(O,A^k;k) = \mathcal{E}_{inv}(TO,A^k;k)
$$

$$
T\mathcal{E}_{equiv}(O,A^k;k) = \mathcal{E}_{equiv}(TO,A^k;k)
$$

(21)

We firstly use $\mathcal{E}_{inv}$ to denoise $K-1$ steps, so for any $1 < k \le K$, the denoise iteration satisfies

$$
\hat{A}^{k\to0} = \mathcal{E}_{inv}(O,A^k;k)
$$

(22)

When the input observation is transformed by $SE(3)$ element, for the property of $\mathcal{E}_{inv}$, we have

$$
\forall_{T\in SE(3)} : \hat{A}^{k\to0} = \mathcal{E}_{inv}(TO,A^k;k)
$$

(23)

It means the predicted noise $\hat{A}^{k\to0}$ keeps invariant no matter what $SE(3)$ transformation is applied on the input observation. And then we carry each element of the predicted noise sequence into 10 , it's obvious that $\mathbf{H}_i^{k-1}$ is also $SE(3)$ invariant for any $1 \le i \le T_p$. So we can infer that $\mathbf{H}_i^1$ is $SE(3)$ invariant to input observation. In terms of Markov transition, the first $K-1$ transitions are $p_1$-like.

$$
p(\mathbf{H}_i^{k-1}|\mathbf{H}_i^k,O) = p(\mathbf{H}_i^{k-1}|\mathbf{H}_i^k,TO), 1 < k \le K
$$

(24)

For the last denoising iteration, we use a $SE(3)$ equivariant model to predict noise, so when the input observation is transformed, we have

$$
\forall_{T\in SE(3)} : T\hat{A}^{1\to0} = \mathcal{E}_{equiv}(TO,A^1;1)
$$

(25)

Carry each element the result into the last denoise step $\mathbf{H}_i^0 = \hat{\mathbf{H}}_i^{1\to0}\mathbf{H}_i^1$, we will discover the final denoised action sequence $\hat{A}^0$ is $SE(3)$ Equivariant. In another word, the last Markov transition is $p_2$-like.

$$
p(\mathbf{H}_i^0|\mathbf{H}_i^1,O) = p(T\mathbf{H}_i^0|\mathbf{H}_i^1,TO)
$$

(26)

Additionally, as the initial noisy action $\mathbf{H}_i^K$ is sampled from Gaussian distribution, it's not conditioned on the observation.

$$
p(\mathbf{H}_i^K|O) = p(\mathbf{H}_i^K|TO)
$$

(27)

Combine 24, 26, 27 together and put them back into 20, we find the whole diffusion process for single action is $SE(3)$ equivariant. Joint all the $\mathbf{H}_i^0(0 \le i \le T_p)$ into a sequence $A^0$, it's easy to verify proposition 2 holds. In other word, **our predicted action sequence is theoretically $SE(3)$ equivariant to input observations**.

# D  EXPERIMENTS SHOWING OUR PROPOSED MARKOV PROCESS IS EASIER TO LEARN

The properties of three different Markov transitions can be described as

$$
\begin{aligned}
p(y|x,c) &= p_1(y|x,Tc) \\
p(y|x,c) &= p_2(Ty|x,Tc) \\
p(y|x,c) &= p_3(Ty|Tx,Tc)
\end{aligned}
\tag{28}
$$

In practice, we use the $SE(3)$ Transformer (Fuchs et al., 2020) with different input and output feature types to approximate the three types of transitions(Denoted as P1Net, P2Net and P3Net).

## D.1  SINGLE-STEP EVALUATION

In this validation experiment, we take a point cloud $P$ with random orientation as observation (focusing solely on rotation for simplicity). The detailed input and output feature types are shown in table 5 . According to the features of $SE(3)$ Transformer, it's easy to verify the networks satisfy the corresponding equivariant properties.

Table 5: Input and Output Feature Types

|        | Input Feature | Output Feature | Supervision       | Final Loss |
|--------|---------------|----------------|-------------------|------------|
| P1Net  | 3 type-0      | 9 type-0       | identity matrix   | 0.0002     |
| P2Net  | 3 type-0      | 3 type-1       | Pose of input pts | 0.25       |
| P3Net  | 1 type-1      | 3 type-1       | Pose of input pts | 0.27       |

For all three networks, the input feature consists of 3 scalar values attached to each point, and the output feature consists of 9 scalar values (after pooling across all points). For P1Net, the output is set as nine type-0 features, meaning the output remains invariant to the rotation of the input point cloud. We supervise the output by computing the L2 loss between it and a fixed rotation matrix. In contrast, for P2Net and P3Net, the output is treated as three type-1 features, which are supervised using the pose of the input point cloud. The only difference among the three networks is the input and output feature types, while all other hyperparameters remain the same.

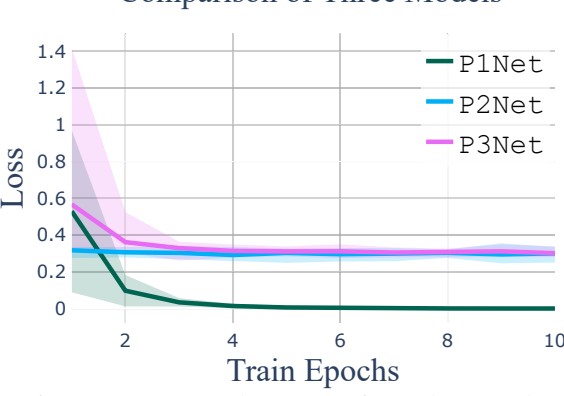

Figure 5: Loss curve of P1Net, P2Net and P3Net. After only several gradient descent, the loss of P1Net converges almost to 0, while the losses of P2Net and P3Net do not decrease obviously.

After training for the same number of epochs, the loss curve of the three networks is shown in fig. 5. The experiments demonstrate that the invariant model (P1Net) is significantly easier to train compared to the equivariant models (P2Net and P3Net), as it is expected to output the same values regardless of the transformation applied to the input point cloud. Additionally, we observe that the use of higher-type features in P2Net and P3Net results in increased memory requirements and longer inference times.

D.2 MULTI-STEP EVALUATION

Moreover, to further demonstrate the superiority of our proposed diffusion process, we compose P1Net, P2Net and P3Net in different ways and evaluate the performance. In this experiment, we compare two diffusion processes, both are end-to-end equivariant to the condition $c$, but have different kinds of transitions. Same as the single-step experiment, we take a point cloud P with random orientation as observation (fo- cusing solely on rotation for simplicity.

For the first process, all transitions are $p_3$-like. So we use one P3Net to approximate the transition function. The input features of this P3Net consists of 3 type-1 features(same size with a rotation matrix), and one type-0 scaler indicating the index of current denoising step. And the output of the P3Net is also 3 type-1 features. So the first diffusion process can be written as

$$x_{k-1} = \text{P3Net}(P, x_k, k), for\ 1 \leq k \leq K \tag{29}$$

And for the second process, the first $K - 1$ denoising steps are $p_1$-like, and the last is $p_2$-like. So here we use one P1Net and one P2Net to approximate the transitions. The input of the P1Net is 10 type-0 features, 9 representing the rotation matrix, 1 representing the index of denoising step, and the output is 9 type-0 features representing the rotation matrix. The input of the P2Net is 9 type-0 features representing the rotation matrix, as we only use it in the last step, we don't input the step index into it. And the P2Net outputs 3 type-1 features. Formally, the diffusion process can be written as

$$x_{k-1} = \text{P1Net}(P, x_k, k), for\ 2 \leq k \leq K \tag{30}$$

$$x_0 = \text{P2Net}(P, x_1) \tag{31}$$

According to the properties of $SE(3)$ Transformer, it's easy to verify the two diffusion processes are both equivariant to the input point cloud $P$. We generate same number of diffusion data pieces $\{P, x^{K:0}\}$ and separately train the models in two diffusion process. The supervision is added on the output of each transition step. To compare the performance of the two diffusion process, we show the final loss $\mathcal{L}(x_{pred}^0, x_{gt}^0)$ of the two process with respect to the training epoch.

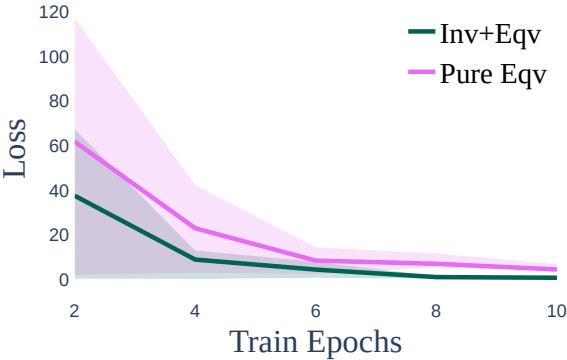

Figure 6: Loss curve of two diffusion process. After only several gradient descent, the loss of Inv.+Eqv. process onverges much faster than the Pure Eqv. process

As shown in fig. 6. , the Inv.+Eqv. process converges much faster than the Pure Eqv. process.

With the two experiments, we demonstrate our choice of using $K - 1$ $p_1$-like denoising steps and only one $p_2$-like step is better than using $K$ $p_3$-like steps.

# E IMPLEMENTATION OF $SE(3)$ EQUIVARIANT AND INVARIANT BACKBONES

Here we introduce the implementation of our true $SE(3)$ equivariant backbone $\mathcal{E}_{equiv}$ and invariant backbone $\mathcal{E}_{inv}$ $SE(3)$ Transformer (Fuchs et al., 2020).

In general, each module consists of 2 $SE(3)$ Transformers, called as *pos_net* and *ori_net*, outputting translation and rotation separately. As the output of $SE(3)$ Transformer is per-point features, we implement a mean pooling over all points to get global features.

### E.1 INVARIANT MODULE

We set the output of *ori_net* as $6 \times T_p$ type0 features, and then implement Schimidt orthogonalization to get rotation matrix. As the $6 \times T_p$ type0 features are $SE(3)$ invariant, the rotation matrix is also invariant.

We set the output of *pos_net* as $3 \times T_p$ type0 features, which is naturally invariant to any SE(3) transformation of the point cloud, guaranteed by the translation invariance of $SE(3)$ Transformer.

Finally we combine each translation and rotation parts to a $4 \times 4$ matrix. So we get $T_p$ of $4 \times 4$ matrices, and all of them are invariant to any $SE(3)$ transformations of input point cloud.

### E.2 EQUIVARIANT MODULE

We set the output of *ori_net* as $2 \times T_p$ type1 features, and then implement Schimidt orthogonalization on each 2 type1 features to get $T_p$ of rotation matrices. As the $2 \times T_p$ type1 features are SE(3) equivariant, the rotation matrices are also equivariant.

We set the output of *pos_net* as $2 \times T_p$ type1 feature and $3 \times T_p$ type0 feature(denoted as *offset*). First we implement Schimidt orthogonalization on each 2 type1 features, get a rotation matrix (denoted as $R$). Additionally, we denote the mass center of the input point cloud as $\mathcal{M} := \frac{1}{N}\Sigma_{i=1}^{N}x_i$, $x_i$ is the coordinate of the $i$-th point. Then we can write each of the predicted translations $t$ as

$$t(\mathcal{X}) = \mathcal{M} + R \cdot \textit{offset} \tag{32}$$

We can prove this translation vector is equivariant to any $SE(3)$ transformation of the input point-cloud $\mathcal{X}$. When the input point cloud is transformed, $\mathcal{X}' = R_{data}\mathcal{X} + t_{data}$.

$$
\begin{aligned}
t(\mathcal{X}') &= (R_{data}\mathcal{M} + t_{data}) + R_{data}R \cdot \textit{offset} \\
&= R_{data}(\mathcal{M} + R \cdot \textit{offset}) + t_{data} \\
&= R_{data}t(\mathcal{X}) + t_{data}
\end{aligned}
\tag{33}
$$

Finally we combine each translation and rotation parts to a $4 \times 4$ matrix. So we get $T_p$ of $4 \times 4$ matrices, and all of them are equivariant to any $SE(3)$ transformations of input point cloud.

## F EQUIVARIANT DIFFUSION ON SE(3) MANIFOLD

The noisy action of step $k$ can be represented as

$$\mathbf{H}^k = \underbrace{\text{Exp}\left(\gamma\sqrt{1-\bar{\alpha}_t}\varepsilon\right)}_{\text{Perturbation}} \underbrace{\mathcal{F}\left(\sqrt{\bar{\alpha}_t}; \mathbf{H}^0, \mathbb{H}\right)}_{\text{Interpolation}}, \varepsilon \sim \mathcal{N}(\mathbf{0}, \mathbf{I}) \tag{34}$$

The first term, $\text{Exp}(\gamma\sqrt{1-\bar{\alpha}_t}\varepsilon)$ is a random noise on $SE(3)$ manifold, aiming to randomize the diffusion process. According to the Gaussian distribution on $SE(3)$ (appendix A.3), a $SE(3)$ Gaussian variable can be written as the Exp of a Gaussian variable on $\mathfrak{se}(3)$ . So we first randomly sample a 6D noise $\varepsilon \in \mathbb{R}^6$ from unit Gaussian distribution, then scale it by a scheduler factor $\gamma\sqrt{1-\hat{\alpha}_t}$ to control the magnitude of the perturbation at different steps. Finally we use the Exp map to convert the variable back to $SE(3)$ .

The second term is an interpolation on $SE(3)$ manifold between $\mathbf{H}^0$ and $\mathbb{H}$. The idea behind this function is, first project the $SE(3)$ transformation to $\mathfrak{se}(3)$ , perform linear interpolation in this tangent space, and then convert the interpolated vector back to $SE(3)$ to obtain the interpolated transformation. One can refer to Jiang et al. (2024) for more details. Formally, the interpolation funcion $\mathcal{F}$ can be expressed as

$$\mathcal{F}\left(\sqrt{\bar{\alpha}_t}; \mathbf{H}_0, \mathbb{H}\right) = \text{Exp}\left((1-\sqrt{\bar{\alpha}_t}) \cdot \log\left(\mathbb{H}\mathbf{H}_0^{-1}\right)\right)\mathbf{H}_0 \tag{35}$$

## G    SIMULATION TASKS– FURTHER DETAILS

- *Rotate Triangle:* A robotic arm with 2D anchor pushes the triangle to a target 6D pose. The task reward is computed as the percentage of the Triangle shape that overlaps with the target Triangle pose.

- *Open Bottle Cap:* A bottle with a cap is placed at a random position in Workspace, and a robot arm is tasked with opening the cap. In this task, the demonstrations show robots Unscrewing bottle cap with parallel gripper. Success in this task depends on whether the bottle cap is successfully opened. Note that, due to simulator constraints, opening the bottle cap simply involves lifting it upward without the need to twist it first.

- *Open Door:* This task evaluates the manipulation of articulated objects. The model is required to generate trajectories to open doors positioned at various orientations. The demonstration is given as: We initialize the gripper at a point $p$ sampled on the handle of the door and set the forward orientation along the negative direction of the surface normal at $p$. And then we pull the door by a degree. Different from door pushing, we perform a grasping at contact point $p$ for pulling. Success in this task is determined by the opening angle of the door.

- *Robot Calligraphy:* This long-horizon task involves using a robot arm to write complex Chinese characters on paper, accounting for different orientations. Success in this task is determined by the aesthetic quality and accuracy of the characters or patterns formed, which should closely resemble the target trajectory.

- *Fold Garment:* A long-horizon task involving deformable object manipulation, where a robot folds a long-sleeved garment. The robot folds the sleeves inward along the garment's central axis, then gathers the lower edge of the garment and folds it upward, aligning it with the underarm region. A folding succeeds when the Intersection-over-Union (IOU) between the target and the folded garments exceeds a bar (Xue et al., 2023a; Canberk et al., 2022).

- *Fling Garment:* A dual-arm task for manipulating deformable objects. The robot grasps the two shoulder sections of a wrinkled dress, lifting it to allow the fabric is clear of the surface. Then flings the garment it to flatten the fabric, and then places it back onto a flat surface. Success is determined by the projection area of the flattened garment.

Some of the six tasks are exactly $SE(3)$ equivariant, and some are partially. In the *Open Door* task, the manipulation trajectory is exactly equivariant with the point cloud of the door. In the *Robot Calligraphy* task, the manipulation trajectory is exactly equivariant with the point cloud of the paper and the handwriting that has been written. In the *Open Bottle Cap* task, the manipulation trajectory is exactly equivariant with the point cloud of the bottle. In the *Rotate Triangle* task, as we always add same transformation on initial pose and target pose of the triangle, the manipulation trajectory is exactly equivariant with the point cloud of the triangle. In the *Fling Garment* and *Fold Garment* tasks, the trajectories are not exactly equivariant with the initial observation of the garment, as the deformation differs in each initialization. But even so, our method can also outperform baselines.

The intuition is, by ensuring end-to-end $SE(3)$ equivariance, our model treats a point cloud in the observation space and all the point clouds possible by $SE(3)$ transformation as an equivalence class, so the observation space is reduced to the quotient space of the observation space over the $SE(3)$ group. Once the $SE(3)$ equivariance is naturally ensured, the network can focus on the geometric features of the object, so it still has the ability to generalize to the deformation and geometry of objects.

## H    REAL TASKS– FURTHER DETAILS

In the task of **Open Bottle Cap**, unlike in a simulator, the process in the real world involves first twisting the cap and then lifting it off to open. The bottle is initially placed at a random position on the table. For **Opening Door**, the initial position of the cabinet is randomly determined. In **Calligraphy**, we use flat-bristled brushes and watercolor paper, which is randomly positioned on the table. In **Cloth Folding**, the limited workspace of the Franka robot means it cannot reach every possible location. Therefore, the placement of clothes is not random but is instead based on locations accessible to the robot.

