# OpenReview forum: "ET-SEED: EFFICIENT TRAJECTORY-LEVEL SE(3) EQUIVARIANT DIFFUSION POLICY"
_ICLR.cc/2025/Conference — ICLR 2025 Poster_

### Official Review · Reviewer_kmzf · 2024-11-02

**Soundness:** 2
**Presentation:** 3
**Contribution:** 4
**Rating:** 8
**Confidence:** 4

**Summary:**

The paper proposes a novel decomposition of an SE(3)-equivariant Markovian denoising process, classifying each denoising step into $p_1$, $p_2$, and $p_3$-type equivariance. In contrast to previous approaches that utilize $p_3$-type equivariance in which the denoising is equivariant to both the context input $c$ and the noised sample $x^{k}$, the authors demonstrate that equivariant denoising can be achieved through a combination of $p_1$ and $p_2$-type equivariance. Furthermore, the authors argue that training $p_1$ models is significantly easier than training $p_2$ and $p_3$ models, as $p_1$ denoising is invariant to $c$, whereas $p_2$ and $p_3$ denoising are equivariant to it. Consequently, the proposed ET-SEED model employs the invariant $p_1$ for all denoising steps except the final step, which uses $p_2$.

**Strengths:**

### **Strength 1. Novelty**
The paper introduces a novel approach to the SE(3)-equivariant pose/trajectory generation problem. Unlike previous works, ET-SEED achieves full equivariance by first generating the trajectory independently of the reference frame in which the context input (e.g., point cloud) is observed, then subsequently transforming it equivariantly with respective to the context frame. Results in Appendix E indicate potential advantages of this approach, though this conclusion should be interpreted with caution, as the experiment in Appendix E is limited to a single timestep and does not account for potential interactions across different timesteps.

Overall, the paper offers valuable new insights into defining equivariance condition for diffusion models, making it worth sharing with the community regardless of its limitations.

### **Strength 2. Experimental Results**
The authors present a benchmark comparison against two state-of-the-art methods (DP3 and EquiBot) across six simulated and four real-world tasks. ET-SEED outperforms these methods by a substantial margin in most settings, clearly demonstrating the benefits of this approach. It is particularly exciting to observe robots learning to perform complex, trajectory-level real-world tasks with only 20 demonstrations.

**Weaknesses:**

### **Weakness 1. No interdependence between each keypose**
The primary limitation of the proposed ET-SEED is that each keypose is denoised independently, with no interaction term between them. Consequently, this is not a true trajectory-level generative model. Joint modeling of all keyposes within a trajectory is indispensable to recent successes in trajectory-level generative models such as diffusion policy and action chunking transformer. Assuming each keypose is independent from the others is unrealistic.

For example, if the robot needs to open a bottle cap and a drawer, it could start by grasping either the bottle cap or the drawer handle. However, once it has selected one action, say, grasping the bottle cap, the subsequent step should be opening the bottle cap rather than the drawer. While this issue could be addressed in a closed-loop manner by putting the current state as input, this approach is less suitable from an open-loop planning perspective.

I am willing to raise my score if this is my misunderstanding. Otherwise, I cannot give a very high score to the soundness of the method.


### **Weakness 2. Experiment in Appendix E**
The experiments in Appendix E provide a compelling reason for preferring $p_1$-type over $p_3$-type denoising. However, the comparison is limited, as each model is evaluated only with a single-timestep kernel, ignoring potential positive interactions that might arise across timesteps with $p_3$-type denoising.


### **Weakness 3. Requirement for Segmentation**
Many recent equivariant robotic manipulation models exhibit strong robustness to unsegmented inputs. In contrast, ET-SEED relies on object segmentation. While segmenting target objects is not particularly difficult these days due to open-vocab segmentation models like SAM, this requirement restricts ET-SEED’s applicability to object-centric tasks. With segmentation, the model is unable to manipulate non-object entities (e.g., a pile of tiny objects as a whole) or understand global scene context.

**Questions:**

Suggestion: Why did you use the SE(3)-transformer? More advanced models, like Equiformer v2, with improved efficiency and scalability are available nowadays.

---

> ### Author Response · Authors · 2024-11-20
>
> We sincerely appreciate Reviewer kmzf’s recognition and positive feedback on our work, as well as the valuable comments and suggestions. Below, we provide detailed responses to the issues raised:
>
> ### Interdependence Between Keyposes
> We apologize for not sufficiently emphasizing this aspect in the original manuscript. In fact, our method, similar to diffusion policy, operates as a true trajectory-level generative model. Specifically, in our implementation, the input to the denoising network is a noisy action sequence $A^k$, and the output is the predicted noise $\hat{A}^{k \rightarrow 0}$ for that sequence. The keyposes within the sequence are not independent; during the forward pass, they interact and influence each other. We regret any misunderstanding caused by our unclear writing.  To address this, we have clarified this point and refined the relevant formulations, particularly in Section 4.3 of the revised manuscript. We hope these revisions offer a clearer understanding of our approach.
> ### Multi-Step Verification of Equivariant Markov Processes
> As you correctly pointed out, single-step experiments alone are insufficient to fully validate the benefits of our diffusion process design. Following your suggestion, we have designed multi-step experiments to analyze the effects of different transition function designs on the performance of the diffusion process. Specifically, we compared two types of diffusion processes:
>
> (1) A process where all steps use $p_3$​-type transition functions (aligned with our baseline EquiBot).
>
>  (2) A process where the first $K-1$ steps use $p_1$-type transition functions and the final step uses a $p_2$-type transition function (the design used in this work).
>
> Experimental results show that the second design outperforms the first in both convergence speed and final performance. The details of these experiments are provided in Appendix D of the revised manuscript, with the newly added second experiment highlighted in red for emphasis. We hope these additional experiments address your concerns and further validate our design choices.
>
> ### Dependency on Segmentation
> Your understanding is entirely correct: ET-SEED requires segmenting the scene’s point cloud to isolate the points corresponding to the target object. In our experiments, we used the SAM2 model for preprocessing the point clouds captured by the cameras. It is worth noting that many prior works leveraging equivariance for pose estimation or trajectory generation assume access to pre-segmented point clouds$^{[1,2]}$. While the dependency on segmentation is indeed a limitation of our method, as you pointed out, recent advancements in open-vocabulary point cloud segmentation models suggest that isolating the target object from a 3D scene is becoming increasingly feasible. With further progress in point cloud segmentation models, challenges like segmenting non-object entities are also expected to be addressed effectively. To acknowledge this limitation, we have added a "Limitations" section in the last of the revised manuscript, highlighted in red.
> ### Selection of Equiformer
> Equiformer V2 is primarily designed with input and structural configurations optimized for atomistic systems, rather than point clouds. Adapting its input design and architecture to effectively handle point clouds is non-trivial, requiring significant modifications to both its data representation and computational framework. To the best of our knowledge, no open-source implementations of Equiformer V2 currently support point clouds as input. Consequently, we have opted to use the SE(3) Transformer as our backbone. In the future, should an open-source implementation of Equiformer V2 for point clouds become available, we plan to integrate it as a replacement for our current backbone, with the expectation of achieving enhanced performance.
>
> We deeply appreciate your thorough review and constructive comments, which have been invaluable in refining the paper's focus and presentation. We have carefully revised the manuscript to address the concerns that lead to your lowered score, and we sincerely hope the changes will allow for a more favorable consideration. If you have any additional questions or concerns, please feel free to reach out to us. Thank you again for your thoughtful review!
>
> [1] Yang, J., Cao, Z. A., Deng, C., Antonova, R., Song, S., & Bohg, J. (2024). Equibot: Sim (3)-equivariant diffusion policy for generalizable and data efficient learning. arXiv preprint arXiv:2407.01479.
>
> [2] Hu, B., Zhu, X., Wang, D., Dong, Z., Huang, H., Wang, C., ... & Platt, R. (2024). OrbitGrasp: $ SE (3) $-Equivariant Grasp Learning. arXiv preprint arXiv:2407.03531.

---

> > ### Author Response · Authors · 2024-11-22
> > **Request for your review**
> >
> > Dear reviewer kmzf,
> >
> > We hope this message finds you well. As we posted our rebuttal to your official review for about 2 days, we kindly request your consideration of our responses to your concerns.
> >
> > We deeply thank you for your time and reviews. We have addressed many of the comments in your review in the revised version, as hopefully you can agree as mentioned above. We thank you for taking the concern to provide critiques of the paper, and hope that our clarifications sufficiently address your concerns and improve your ratings.
> >
> > Thank you for your attention and consideration.
> >
> > Authors

---

> ### Comment · Reviewer_kmzf · 2024-11-26
>
> Dear authors,
>
> Thank you for clarifying the keypose interdependence and adding a comparison experiment in Appendix D.2. All of my concerns have been addressed, and I have raised my score and confidence accordingly.

---

### Official Review · Reviewer_eYAU · 2024-11-04

**Soundness:** 3
**Presentation:** 3
**Contribution:** 3
**Rating:** 8
**Confidence:** 3

**Summary:**

This paper proposes a new method for SE(3) equivariant diffusion policy. The key is a proved theorem that in the denoising process, only the last denoising step needs to be equivariant, and all previous denoising steps can be invariant. A new diffusion policy is designed based on this results, where the first K-1 denoising steps employ a invariant transformer, and the last one employs a equivariant transformer. Results on 6 simulation tasks show the proposed method, ET-SEED, performs better than existing baselines, especially on new poses not seen during training.

**Strengths:**

- This paper is overall very clearly written.
- The proposed method is straightforward in implementation yet very effective.
- The experiments results show the strong performance of the proposed method.

**Weaknesses:**

- Overall I think this is a good paper and do not have major weaknesses. One minor comment: for figure 3, the block for Inv. SE(3) transformer and Eqv. SE(3) transformer are identical, which seems like a typo?

**Questions:**

see weakness section

---

> ### Author Response · Authors · 2024-11-17
>
> We sincerely appreciate your positive recognition of our work and are grateful for your constructive feedback. Below is our response to your comment:
>
> The Inv. SE(3) Transformer and Eqv. SE(3) Transformer indeed share the same network architecture. The key distinction lies in the types of input and output features they process. As you correctly pointed out, the original representation in Figure 3 might lead to some confusion. To address this, we have added an explanatory note in the figure caption(highlight with red font) to clarify the differences between the two models, while also referencing the detailed explanation provided in the appendix.
>
> Thank you once again for your insightful suggestions, which have highlighted important areas for improvement. Should you have any further questions or suggestions, please feel free to reach out to us.

---

### Official Review · Reviewer_t1h8 · 2024-11-04

**Soundness:** 3
**Presentation:** 3
**Contribution:** 3
**Rating:** 6
**Confidence:** 5

**Summary:**

This paper combines score-based modeling and geometric deep learning for robot policy learning. The results are evaluated in both simulated and real-world robot environments. The authors claim that the proposed approach attains stronger performance than existing baselines.

**Strengths:**

- Real-world experiments are valuable. I appreciate the authors' efforts to conduct real-world validations of their approach
- Baseline comparisons add value. I appreciate the authors' efforts to compare against 3D representation policy learning works and probabilistic modeling of multi-modal trajectories such as EquiBot.

**Weaknesses:**

**A. Presentation**.

The presentation of the paper needs a lot of care in both writing and figures.

There are currently too many display items in the teaser figure, some of which distract the readers from the fruit of the proposed technique. If the authors aim to convince the audience of probabilistic modeling of trajectories and geometric learning, then the figure should focus on those two things. Showing the performance graph directly in the first figure does not clarify that point. I would refer the authors to the teaser figures in diffusion policy and vector neurons, both of which are cited by the authors. Currently, the teaser figure does not strongly convey how trajectories are equivariant and how multi-modality is captured by score-based modeling. Perhaps part of the Figure 2 design can be used to improve this.

A few minor points: I am also unsure if you need almost ten lines of text to clarify the definitions of equivariant and invariant functions (L162-172). I'm not sure whether Algorithm and Figure 3 need to be taking that much space. Perhaps you can make more space there and add my suggested experiments.

**B. Missing critical experiments, novelty concerns**.

I am currently unconvinced and cannot fully assess the performance of the score-based modeling and geometric modeling. If the goal of this work is to showcase the capabilities of combining score-based probabilistic modeling and geometric learning, then I think the two strongest values would be (1) handling multi-modality in the supervision signals for imitation learning, as shown in many works such as diffusion policy, and (2) equivariant representations. If these are the authors' aims, these two points must be substantiated more deeply with visualizations.

I cannot find a single visualization of the generated trajectory in the main manuscript that showcases multi-modality. I cannot find a visualization that showcases equivariant representation either. These two points are not substantiated.

This is deeply related to the novelty of the work. By just reporting on the criterion of success rate, it is truly difficult to understand what caused the performance gaps between the prior works and this one. The dynamical process is abstracted away into a single scalar evaluation criterion. I need more convincing visualizations and results to understand and assess this work. The leading performance reported by the authors could be due to a bunch of things, and whether it is related to the incorporation of probabilistic and geometric modeling is hard to assess.


**C. Implicit Assumptions, evaluation criteria**.

There seem to be several assumptions in this work that I encourage the authors to clarify, perhaps in a limitation section. For example, there seem to be assumptions about perception and policy representation. The perception part needs to be a point cloud and a specified coordinate system by experts. The policy part assumes 6 DOF end-effector representation and the availability of an inverse kinematics/dynamics controller. These assumptions are non-trivial when it comes to more dexterous tasks. Consider spinning a pen with a robot hand. Do we expect the commands of the robot hand to be specified in task space, which would be highly difficult for high-frequency control, and what is the notion of equivariance there?

The presented tasks seem to be primarily trajectory generations in a single Euclidean frame. If this is true, it needs to be specified in the paper, perhaps a limitation section.

**Questions:**

Please see my comments in the weakness section. Thanks.

---

> ### Author Response · Authors · 2024-11-25
> **Thank you for the valuable review (1/2)**
>
> We sincerely thank Reviewer t1h8 for your insightful comments and suggestions. Below, we provide detailed responses to the points raised.
>
> ### Core Contribution
> We would like to take this opportunity to clarify the core contribution of our work: introducing an SE(3) equivariant diffusion model that leverages spatial symmetries to improve data efficiency and spatial generalization in robotic manipulation tasks while reducing training complexity. Combining score-based modeling and geometric learning has been demonstrated in prior works as an effective approach for generating robot manipulation poses and trajectories. By leveraging the inherent symmetry in manipulation tasks, such methods significantly reduce the number of required demonstrations and improve spatial generalization capabilities [1–4]. Furthermore, this kind of approach has garnered considerable attention from the community. For instance, Equivariant Diffusion Policy [4] was nominated for the Best Paper award at CoRL 2024.
>
> While we are not the first to integrate score-based modeling with geometric learning, our key contribution lies in introducing a novel end-to-end SE(3)-equivariant diffusion policy. Unlike previous methods, our approach optimizes the choice of the transition function in the diffusion process and directly defines the diffusion variables on the SE(3) manifold rather than in Euclidean space. These innovations enable our method to maintain end-to-end equivariance while achieving superior data efficiency and spatial generalization.
>
> ### Teaser and Visualization of Multimodal Trajectories
> We appreciate your feedback regarding the teaser figure. We agree that directly including performance graphs in the teaser is not an ideal approach. Based on your suggestion, we have updated the teaser by adding a set of images(Fig.1(b)) illustrating how our method generates equivariant manipulation trajectories (as our actions are defined as the 6D poses of the end-effector, so the manipulation trajectory is exactly the equivariant representations you were concerned about). Additionally, we showcase its ability to handle multimodal action distributions.
> To demonstrate this, we selected a fold garment task where multimodality is evident. For the same piece of cloth, one can fold either the left sleeve or the right sleeve. Both modalities are present in the demonstrations. During testing, when given a cloth placed in an arbitrary pose, our method randomly selects one of the two modes to complete the folding task. This example highlights both our method’s ability to handle multimodal task distributions and its spatial generalization capability.
> We have also revised Fig. 2 to more clearly showcase our core contribution: a more efficient SE(3)-equivariant diffusion policy. Additionally, we addressed the minor points you raised. Specifically, we removed the separate box defining equivariant and invariant functions(L150-L152) to create space for new experiments, and we improved the layout of Fig. 3 and the pseudocode for the algorithm.
> ### Evaluation Criterion
> In our paper, we evaluate our method and the baselines using two metrics: the success rate of manipulation tasks and the geodesic distance between the predicted trajectories and ground-truth trajectories. We explicitly point out in the paper (L437-L449,the font is marked in red) that success rate alone is insufficient for evaluating trajectory quality. This is why we additionally report the geodesic distance, which you may have overlooked.
> ### Visualization of Equivariant Trajectories
> We agree with your suggestion to include additional visualizations of the generated manipulation trajectories. We have added a new figure (Fig. 5) illustrating the trajectories generated by our method compared to those of DP3 for one of the tasks in our experiments (fling garment). As shown in the figure, when the cloth is placed at a new pose, our method can generate an effective manipulation trajectory, while DP3 generates inaccurate end-effector poses in the first several steps, resulting in the failure of the entire manipulation process.  You can find this figure in the revised version of the paper(around L410), with captions in red font.

---

> ### Author Response · Authors · 2024-11-25
> **Thank you for the valuable review (2/2)**
>
> ### Limitations
> We acknowledge that our method relies on certain assumptions. As you mentioned in the comment, we define the action space as the 6D pose of the end-effector, similar to the formulation of Diffusion Policy$[5]$ and EquiBot$[1]$. We added a “Limitation” section in the last of the revision version, in order to point out these assumptions.
>
> However, we would like to clarify the statement in Weakness.C: "The perception part needs to be a point cloud and a specified coordinate system by experts." This statement is not entirely accurate. Our method does not require a “specified coordinate system by experts.” As outlined in Definition 1, we only assume the existence of a coordinate system attached to the object, but no specific information about it is necessary.
>
> ### Summary
>
> We deeply appreciate your review and constructive comments, which have been invaluable in refining the paper's focus and presentation. We have carefully revised the manuscript to address the concerns that lead to your lowered score, and we sincerely hope the changes will allow for a more favorable consideration. If you have any additional questions or concerns, please feel free to reach out to us. Thank you again for your thoughtful review!
>
> [1] Yang, J., Cao, Z. A., Deng, C., Antonova, R., Song, S., & Bohg, J. (2024). Equibot: Sim (3)-equivariant diffusion policy for generalizable and data efficient learning. arXiv preprint arXiv:2407.01479.
>
> [2] Hu, B., Zhu, X., Wang, D., Dong, Z., Huang, H., Wang, C., ... & Platt, R. (2024). OrbitGrasp: $ SE (3) $-Equivariant Grasp Learning. arXiv preprint arXiv:2407.03531.
>
> [3] Gao, C., Xue, Z., Deng, S., Liang, T., Yang, S., Shao, L., & Xu, H. (2024). RiEMann: Near Real-Time SE (3)-Equivariant Robot Manipulation without Point Cloud Segmentation. arXiv preprint arXiv:2403.19460.
>
> [4] Wang, D., Hart, S., Surovik, D., Kelestemur, T., Huang, H., Zhao, H., ... & Platt, R. (2024). Equivariant diffusion policy. arXiv preprint arXiv:2407.01812.
>
> [5] Chi, C., Xu, Z., Feng, S., Cousineau, E., Du, Y., Burchfiel, B., ... & Song, S. (2023). Diffusion policy: Visuomotor policy learning via action diffusion. The International Journal of Robotics Research, 02783649241273668.

---

> > ### Comment · Reviewer_t1h8 · 2024-11-26
> > **Thank you for the rebuttal**
> >
> > I thank the authors for their rebuttal efforts. I have thus increased my score.

---

### Official Review · Reviewer_fnNr · 2024-11-05

**Soundness:** 4
**Presentation:** 4
**Contribution:** 4
**Rating:** 6
**Confidence:** 3

**Summary:**

This paper presents a framework for learning SE(3) equivariant policies with diffusion models. The training and inference of the diffusion model is based on invariance and the SE(3) equivariance properties. Both simulation and real-world experiments are conducted and show promising results on four robotic tasks.

**Strengths:**

- The framework presented in this paper is interesting. Incorporating equivariance property in diffusion policy makes sense.
- The presentation of the paper is good.
- The experiments conducted in this paper is thorough and the results are convincing.

**Weaknesses:**

- Not sure if including too much math background in the appendix is appropriate. Most content in Sections A and B, if not all of, is unrelated to this paper and should be deleted. Please don't expect that including seemingly sophisticated math in the paper can make the paper look awesome superficially and thus get better rating scores from reviewers. I actually lowered my score because of these unrelated math being included.
- Some math notations are not defined. For example, $E_{equiv}$ and $E_{inv}$.

**Questions:**

- What is exactly the format of observation? Is it RGB images? Or colored 3D point clouds? How did the authors guarantee the transformation of input observation T is a simple transformation so that they can verify the equivariance of the policy with T?
- In Algorithm 1, I didn't see how the equivariance and invariance are included anywhere. Does it mean that equivariance are only guaranteed because of inference?
- How are $E_{equiv}$ and $E_{inv}$ defined?

---

> ### Author Response · Authors · 2024-11-17
>
> We sincerely appreciate your positive evaluation of our paper and your valuable feedback. We have carefully addressed the concerns you raised, as detailed below:
>
> - On the mathematical background in Appendix A and B:
>
> We originally included detailed mathematical background in Appendix A and B to provide readers with comprehensive context. However, as you correctly pointed out, such extensive details may not be necessary for the paper's core contributions. In the updated version, we have significantly streamlined Appendix A, retaining only the most relevant mathematical background directly tied to our work. Additionally, we have cited a mathematics textbook to guide readers interested in further exploration of this topic.
>
> - On the definitions of $E_{equiv}$ and $E_{inv}$​:
>
> We apologize for the typo in Equation (5) in the original manuscript, where the left-hand side should have been $E_{inv}$. This has been corrected in the revised version. Both $E_{inv}$ and $E_{equiv}$ represent our modified models based on the SE(3) Transformer, with invariance and equivariance properties under SE(3) transformations, respectively (refer to Equations (4) and (5)). Additional implementation details for these models have been included in the updated Appendix E.
>
> - On the format of observations and their equivariance:
>
> As mentioned in the "Problem Formulation" part of the Method (Section 4), our observations consist of a colored point cloud. We assume the point cloud only contains the object to be manipulated (in our experiments, we preprocess the point cloud to segment object points from the scene). Under this assumption, a change in the spatial pose of the object corresponds to an SE(3) transformation of the input point cloud. This allows us to use a policy equivariant to spatial transformations to handle such inputs effectively.
>
> - On equivariance and invariance in Algorithm 1:
>
> The equivariance and invariance properties of our model are inherently guaranteed by the network architecture. Thus, during the training phase, no special design is required. We simply follow a DDPM-like supervised learning approach, training the model to predict noise as closely as possible to the actual added noise. As you noted, our method's equivariance is guaranteed during the inference phase, where it outputs appropriate action sequences for objects in varying spatial configurations.
>
>
> We deeply appreciate your thorough review and constructive comments, which have been invaluable in refining the paper's focus and presentation. We have carefully revised the manuscript to address the concerns that lead to your lowered score, and we sincerely hope the changes will allow for a more favorable consideration. If you have any additional questions or concerns, please feel free to reach out to us. Thank you again for your thoughtful review!

---

> > ### Author Response · Authors · 2024-11-22
> > **Request for your review**
> >
> > Dear reviewer fnNr,
> >
> > We hope this message finds you well. As we posted our rebuttal to your official review for about 5 days, we kindly request your consideration of our responses to your concerns.
> >
> > We deeply thank you for your time and reviews. We have addressed many of the comments in your review in the revised version, as hopefully you can agree as mentioned above. We thank you for taking the concern to provide critiques of the paper, and hope that our clarifications sufficiently address your concerns and improve your ratings.
> >
> > Thank you for your attention and consideration.
> >
> > Authors

---

> ### Author Response · Authors · 2024-11-20
> **Request for your review**
>
> Dear reviewer fnNr,
>
> We hope this message finds you well. As we posted our rebuttal to your official review for about 3 days, we kindly request your consideration of our responses to your concerns.
>
> We deeply thank you for your time and reviews. We have addressed many of the comments in your review in the revised version, as hopefully you can agree as mentioned above. We thank you for taking the concern to provide critiques of the paper, and hope that our clarifications sufficiently address your concerns and improve your ratings.
>
> Thank you for your attention and consideration.
>
> Authors

---

> ### Author Response · Authors · 2024-11-27
> **Request for your review**
>
> Dear reviewer fnNr,
>
> We hope this message finds you well. As we posted our rebuttal to your official review for about 10 days, we kindly request your consideration of our responses to your concerns.
>
> We deeply thank you for your time and reviews. We have addressed many of the comments in your review in the revised version, as hopefully you can agree as mentioned above. We thank you for taking the concern to provide critiques of the paper, and hope that our clarifications sufficiently address your concerns and improve your ratings.
>
> Thank you for your attention and consideration.
>
> Authors

---

### Author Response · Authors · 2024-11-25
**Manuscript Update Note**

We thank all reviewers for their invaluable feedback. We have carefully considered their suggestions, addressed their concerns, and revised our manuscript accordingly. Major changes are highlighted in **red** in the revised manuscript. The details of these modifications are as follows:

- We removed excessive mathematical derivations from Appendix A of the initial manuscript, leaving only the parts most relevant to the main theme of the paper (based on Reviewer fnNr's feedback).
- We revised the teaser by removing the performance bar chart and adding visualizations of multimodal action sequences and equivariant trajectories (based on Reviewer t1h8's feedback).
- We redraw Fig. 2 to more clearly illustrate the diffusion process we proposed (based on Reviewer t1h8's feedback).
- We added Fig.5  to visualize the trajectories generated by our method and DP3 for fling garment task, providing a clearer illustration of our method’s performance (based on Reviewer t1h8's feedback).
- We added a Limitation section at the end of the paper to discuss the assumptions inherent in our method (based on Reviewers t1h8 and kmzf's feedback).
- We updated the caption of Fig. 3 to include explanations of the SE(3)-equivariant and invariant transformer (based on Reviewer eYAU's feedback).
- We revised the mathematical derivations in Section 4.3 and the pseudocode to more explicitly indicate that our approach is a true trajectory-level generative model (based on Reviewer kmzf's feedback).
- We included an additional comparison in the appendix D to analyze the training difficulty of diffusion processes composed of different transition functions, further validating the importance of key design choices in our diffusion process (based on Reviewer kmzf's feedback).

We once again sincerely thank all the reviewers for their valuable feedback on our paper. We hope that these additions and revisions effectively address your concerns. We kindly ask the reviewers to consider these changes and re-evaluate the scores for our paper. If any of our responses are unclear, we would be more than happy to engage further with the reviewers for clarification and discussion.

---

### Meta-Review · Area_Chair_SK6z · 2024-12-18

**Metareview:**

The paper presents an approach for equivariant policy learning based on diffusion policy. The motivation to transfer ideas from equivariant deep networks to policy learning is solid. The reviewers agree that the core ideas in the paper are quite interesting and the experiments well done. Particularly, the experiments on both simulation and real robots are quite convincing. There are some concerns in the evaluations and the limitation of needing segmentations. The rebuttal did a good job in addressing many of these concerns and has resulted in an overall positive opinion of this paper.

**Additional Comments On Reviewer Discussion:**

A couple of reviewers increased their scores after the rebuttal and hence my positive rating for this work.

---

### Decision · Program_Chairs · 2025-01-22

Accept (Poster)